# Development and Research of the Sensitive Element of the MEMS Gyroscope Manufactured Using SOI Technology

**DOI:** 10.3390/mi14040895

**Published:** 2023-04-21

**Authors:** Danil Naumenko, Alexey Tkachenko, Igor Lysenko, Andrey Kovalev

**Affiliations:** Design Center of the Microelectronic Component Base for Artificial Intelligence Systems, Southern Federal University, Taganrog 347922, Russia; dvnaumenko@sfedu.ru (D.N.); ielysenko@sfedu.ru (I.L.); avkovalev@sfedu.ru (A.K.)

**Keywords:** micromechanical systems, MEMS, gyroscope, sensitive element, drive mode, sensitive mode, electrical part, mechanical part, prototype sample, experimental studies

## Abstract

In this article, based on the developed methodology, the stages of designing the sensitive element of a microelectromechanical gyroscope with an open-loop structure are considered. This structure is intended for use in control units for mobile objects such as robots, mobile trolleys, etc. To quickly obtain a ready-made gyroscope, a specialized integrated circuit (SW6111) was selected, for the use of which the electronic part of the sensitive element of the microelectromechanical gyroscope was developed. The mechanical structure was also taken from a simple design. The simulation of the mathematical model was carried out in the MATLAB/Simulink software environment. The mechanical elements and the entire structure were calculated using finite element modeling with ANSYS MultiPhysics CAD tools. The developed sensitive element of the micromechanical gyroscope was manufactured using bulk micromachining technology−silicon-on-insulator−with a structural layer thickness equal to 50 μm. Experimental studies were carried out using a scanning electron microscope and a contact profilometer. Dynamic characteristics were measured using a Polytec MSA-500 microsystem analyzer. The manufactured structure has low topological deviations. Calculations and experiments showed fairly accurate results for the dynamic characteristics, with an error of less than 3% for the first iteration of the design.

## 1. Introduction

Microelectromechanical systems (MEMS), along with nanotechnology, is a promising direction in instrument engineering and is actively developing. MEMS are miniature devices that combine electrical and micromechanical components integrated on a single substrate, manufactured using microelectronic technologies.

MEMS gyroscopes along with MEMS accelerometers are among the most popular MEMS devices, since these elements are the basis for building small-sized free-form inertial navigation systems (INSs), orientation systems and stabilization systems and are also used in mobile devices, toys, etc. [1]. This is due to the main advantages of MEMS, such as low weight, low power consumption, high reliability, resistance to mechanical stress and low cost in mass production [2,3,4,5,6,7,8,9,10,11,12].

According to research by Yole Developpement, the MEMS device market is recovering from the effects of the pandemic and, instead of the projected USD 21 billion, is valued at USD 12.1 billion, the same as in 2016, completely absorbing the growth from 2017 to 2019 [13]. Unfortunately, these figures have nothing to do with domestic realities, since millions of batches of the previously mentioned mobile devices are not produced in Russia, so in each case, we are talking about the development and production of relatively small volumes. According to domestic marketing research conducted by the Russian MEMS Association [14], various inertial sensors and technologies for their design, modeling, production and testing are of the greatest interest to Russian enterprises. The specificity of such interest among customers is its niche nature. In this regard, work by Russian scientists is mostly related to improving the accuracy of measurements [15], resistance to mechanical influences [16], calibration [17] and general issues in the design of inertial sensors [1,18]. Fundamental issues in the development of MEMS sensors are also being studied [19,20,21].

The analysis of scientific and technical works shows a wide variety of designs and operating principles of micromechanical gyroscopes. Individual groups of scientists and enterprises focus their efforts on a certain concept of building a MEMS gyroscope and systematically work to improve its characteristics.

Linear LL-type MEMS gyroscopes are the most successful design and have become the most widespread, which is confirmed by a large number of commercial samples and an abundance of scientific publications on these types of MEMS gyroscopes. Linear architecture has not only not been supplanted by other architecture but also been systematically developing since the very moment of the creation of the first MEMS gyroscope in Draper Laboratory [22]. The use of other MEMS gyroscope architectures together is possible in navigation units to increase accuracy by aggregation. These designs have great potential for improving accuracy characteristics by improving the sensitive element (SE), improving the accuracy of measurements and improving algorithms for processing primary information. Among the ways to improve accuracy characteristics and increase resistance to external influences is the addition of compensating and matching mechanical elements, increasing the number of inertial masses (IMs). When developing the MEMS gyroscope, the choice was made to use a linear architecture for the following reasons:-There are many analogs with good characteristics, which indicates a successful design. The abundance of implemented designs also reduces the risks for development, which is very important when a guaranteed result is required in a specified time.-The studied mathematical apparatus simplifies both development and modeling. This also reduces the development time of the MEMS gyroscope.-There are existing scientific and technical foundations in the scientific groups of the Southern Federal University (Taganrog, Russia) for the development of linear-type vibration MEMS gyroscopes.-It offers the ability to work at atmospheric pressure.-It is a relatively simple and inexpensive manufacturing technology, with the possibility of further localization at enterprises.-There is a potential for further performance improvement by increasing the number of IMs, adding synchronizing elements, expanding the dynamic range, etc.-Ready-made application-specific integrated circuits (ASICs) are available.

Using the example of the MEMS gyroscope Tronics Gypro2300LD “Development Platform for MEMS Inertial Sensors”, as shown in Figure 1, it can be seen that the main parts of the MEMS gyroscope are the SE—(**1**); the ASIC—(**2**); and the package—(**3**) [23].

This article discusses MEMS gyroscopes designed and manufactured using bulk micromachining technology−silicon-on-insulator (SOI) technology−with a structural layer thickness equal to 50 μm. It is worth noting that the majority of Russian- and foreign-produced MEMS gyroscopes of the tactical class and higher are produced using bulk micromachining technology.

The results presented in this article are based on the experience gained in carrying out research work, design development, documentation and the study of prototypes of the SE of the MEMS gyroscope. Recommendations for the development of the design of the SE based on the study of models of the SE using the finite element modeling (FEM) method and the results of experimental studies of prototypes are presented. The issues of ASIC development are not discussed in this paper, although the requirements imposed by the processing chip on the SE of the MEMS gyroscope were taken into account. In addition, in this article, special attention is paid to the study of the topology and dynamic characteristics of the SE at the prototyping stage. A real prototype of the SE was used for research. As a result, there were additional difficulties, since it is difficult to isolate individual parameters for which it is desirable to use test structures. It is also worth noting the great practical value, since in subsequent work, developers can compare their results with the results presented in this article.

The main purpose of this work was to develop and study the design of the SE of a MEMS gyroscope manufactured using bulk micromachining technology (SOI technology) with the possibility of the further localization of production. The study of the parameters of the MEMS gyroscope was carried out in order to verify the adequacy of the developed mathematical models and to verify the accuracy of the analytical and FEM methods used in the design.

The main tasks to be solved in the study were the development of a mathematical model of the SE, which allows the modeling of its dynamic characteristics; the development of the design of the SE, which allows the manufacture of a sensor using bulk micromachining technology (SOI technology); the development of finite element models of the SE and, on the basis of these models, numerical studies of the dynamic characteristics of the SE; and experimental studies of the topology and dynamic characteristics of the prototype of the SE of the MEMS gyroscope.

The research methods in this article are based on analytical methods using theoretical mechanics, the theory of vibrations, the theory of elasticity and the resistance of materials. Numerical methods of computer modeling and methods of FEM were used in the development of the SE. Mathematical modeling and FEM were performed using modern software packages. Experimental methods included optical and scanning electron microscopy (SEM), laser-Doppler vibrometry, stroboscopic video microscopy and contact profilometry.

## 2. Signal Processing Circuit

The signal processing circuit is an integral and very important part of the MEMS gyroscope. To date, there are several versions of the signal processing chip. Depending on the degree of integration, these may be electronic circuits on discrete components, microassemblies, field-programmable gate arrays (FPGAs), and custom-made and semi-custom basic matrix crystals (BMCs), but the most modern approach is the manufacture of an ASIC [24]. Developing and ordering one’s own ASIC is an expensive and time-consuming undertaking. In this regard, within the framework of this study, it was decided to use commercially available ASICs, which provide for adjustments to a specific SE within their specifications.

One of the available platforms for the development of capacitive MEMS sensors is the specialized SWS6111 platform, presented in Figure 2, from Si-Ware Systems [25]. This platform is an integrated ASIC with a computer interface for debugging the SE of the MEMS gyroscope.

The development of the SE of the MEMS gyroscope was carried out starting from the ASIC, which allows the creation of a typical SE. To develop a modern MEMS gyroscope with a modern type of SE, for example, multi-mass designs with frequency modulation, multi-axis SE or SE with advanced accuracy characteristics, customized ASIC development of the SE is required; i.e., the development should come from the SE.

The possible electrode structures of the SE for constructing a MEMS gyroscope on the SWS61111 ASIC are shown in Figure 3. In the diagram in Figure 3a, the SE of a direct measurement MEMS gyroscope is presented and contains two pairs of differential capacitors in the drive mode (DM), one of which (A_A1/A_A2) is responsible for pumping the IM and the second of which (A_D1/A_D2) is connected to capacitance–voltage converters. The sensitive mode (SM) consists of a differential capacitor: B_D1/B_D2.D. The scheme in Figure 3b is distinguished by the presence of an additional pair of differential capacitors for holding the IM in the central position.

This article presents a variant of the development of the SE of the MEMS gyroscope with an open-loop structure.

### 2.1. Scheme of Elastic Suspension of the SE

After determining the type of the MEMS gyroscope SE being developed and its electrode structure, several initial designs of the SE with similar characteristics, manufacturing technology and implementation options were selected. The most suitable design was a single-mass SE with a decoupling frame. There are several options for implementing this scheme.

The most well-known version of the suspension with a decoupling frame was developed in METU (Turkey) [26,27,28,29,30,31]. According to the results of experiments, the measurement range is ±50°/s, the time departure from zero is equal to 14.3°/h and the zero offset is less than 0.1°/s.

The suspension of the specified structure also has 4 separate dividing frames. Due to its simplicity and good characteristics, this design was used in the development of the MEMS gyroscopes at Southern Federal University (Taganrog, Russia) [32]. Subsequently, this design was used as a basis for the development of multi-axis MEMS gyroscopes [33] and modified to a two-mass SE with a synchronizing cross-shaped element [34,35]. The disadvantage of this design is the fastening of elastic suspensions in the center of the IM; thus, with an increase in linear dimensions, the torque of rotation of the IM increases, and at high vibration and shock loads, both the occurrence of noise and the failure of the SE are possible.

In connection with this disadvantage, a design with fastenings at the edges was considered. A variant of this design was described by scientists from UC Irvine (USA). This MEMS gyroscope has a high linearity of measurements, a bandwidth of 50 Hz and a sensitivity of 0.91 mV/°/s with a noise component of 0.25°/s/$\sqrt{Hz}$ [36,37,38,39,40,41,42,43,44,45,46,47,48,49]. This design was made by using bulk micromachining manufacturing technology (SOI technology) from a finished wafer with the thickness of the instrument layer being 100 μm. Perforated suspension elements are caused by the need to etch a layer of silicon oxide ($SiO_{2}$) between the substrate and the instrument layer. A feature of this MEMS gyroscope design is the separation of the decoupling frame into two parts and the arrangement of elastic elements along their edges, which gives additional stiffness to the structure.

The developed topology of the SE of the MEMS gyroscope is shown in Figure 4.

The IM (**1**) oscillates in two planes due to elastic suspensions consisting of U-shaped elastic suspensions (U-beam) (**2**). Vibrations in the mode of movement along the axis are simplified due to an electrostatic drive (**3**) located on both sides of it. It consists of three parts with aluminum contact pads (**4**), so it is possible to remove the signal for phase detection, where one part is used to excite the signal, and the other is used for control. The useful signal arising under the influence of the Coriolis force is extracted from two capacitive plates (**5**). The combs of the capacitive converter are located with an offset and operate in differential mode; when the capacity of one of them increases, the other decreases. This allows greater sensitivity to be achieved.

The developed structure (cross-section) of the SE of the MEMS gyroscope is shown in Figure 5. The entire SE is made on a substrate of glass (**6**), which reduces parasitic capacitances. The movement of the IM (**1**) is provided by cavities (**7**) under the entire area of the moving parts and elastic elements (**8**). The depth of the cavities is about 20 μm, which also helps to reduce parasitic capacitances and damping. The silicon instrument layer (**9**) is bonded to the glass substrate through a layer of silicon dioxide $\left(SiO_{2}\right)$ (**10**). Undesirable movements of these in the vertical plane are limited by the support (**11**) under them. The gap between the IM and the support is 2 μm, which is due to the thickness of the silicon dioxide $\left(SiO_{2}\right)$ layer. A metallization layer (**12**) is applied on top, which serves to form contact pads. The thickness of the layer (**12**) is 0.2 μm.

The developed MEMS gyroscope works as follows:-When a sinusoidal signal is applied to the plates of the electrostatic drive under the influence of electrostatic forces in the gaps of the electrostatic vibration drive, the IM performs reciprocating movements in the direction of the X-axis.-When the base rotates with the angular velocity ω around the axis perpendicular to the plane of the IM, Coriolis forces of inertia arise, acting on the IM and directed along the Y-axis. The amplitude of the oscillations of the IM in the direction of the Y-axis is proportional to the measured angular velocity ω.-An electrical signal proportional to the amplitude of the IM oscillations is extracted from the capacitive displacement converter and then converted by an electronic signal processing circuit in the electronic unit.-The decoupling mechanism also minimizes the impact of manufacturing defects and the resulting anisoelasticity by using independent bends and limited moving electrodes in the excitation and reading modes.

### 2.2. The Electronic Part

As a rule, the development and description of MEMS devices begin with the mechanical part. In our case, the first step of the design was the choice of the ASIC, and a necessary condition for the implementation of this task was the selection of the optimal electrode structure for its implementation.

There are a wide variety of types of actuators used in MEMS devices: electrostatic, electromagnetic, piezoelectric, optical and thermal. Within the framework of this work, it was possible to use only electrostatic actuators and capacitive displacement converters, which is due to both the chosen manufacturing technology and the ASIC used. Electrostatic drives provide acceptable characteristics in terms of control voltage and are characterized by low noise, high dynamic sensitivity and low-temperature drift. For this reason, the electronic part of the SE of the MEMS gyroscope being developed consists of an electrostatic power drive and capacitive displacement transducers.

#### 2.2.1. Capacitive Displacement Converter

When developing the design of the MEMS gyroscope, a circuit with a variable gap was chosen for measuring capacitance, as shown in Figure 6a, which is characterized by high sensitivity. The disadvantage of this scheme, which consists of the nonlinearity of the output characteristic, can be minimized if it works in a linear zone with small movements. The measurement of capacitance involves the stator (**1**) with the contact pad (**3**) and the rotor (**2**). To prevent sticking, elastic locking elements (**4**) are installed on the stator. The image of the capacitive element obtained using SEM is shown in Figure 6b. In Figure 6b, notice the trimmed fingers at the edges: this trick is used to reduce the influence of quadrature interference. Additionally, because of the cut fingers, the capacitance decreases, and the trimming is carried out in such a way as to be a multiple of the entire length of the comb. In this case, the adjustment is 2 fingers.

The initial capacitance can be determined using Equation (Equation 1) [50]:(1)$$C=\epsilon_{0}\epsilon_{r}\frac{S}{d_{0}}$$
where $\epsilon_{0}$ is the relative permittivity of the air space; $\epsilon_{r}$ is the electric constant; *S* is the overlap area of the fingers; and $d_{0}$ is the size of the air gap between the fingers.

The calculated parameters of the capacitive converter of the SE of the MEMS gyroscope being developed are given in Table 1.

When exposed to the Coriolis force, there is a deviation from the initial position by a value of *d*. In this case, the capacitance can be determined using Equation (Equation 2):(2)$$C=\epsilon_{0}\epsilon_{r}\frac{S}{d_{0}-d}$$

The maximum stroke is limited by the initial gap, but this type of converter does not use the full operating amplitude during operation due to the nonlinear characteristic. Figure 7 shows a graph of the displacement of the capacitance lining of 1 μm. As can be seen from the graph, it can be considered linear only in a small area from 0 μm to 0.4 μm, and then the error greatly increases. The final recommended capacity is 2.75 pF.

When the tanks are switched on in differential mode, the value for the tanks doubles. With a deviation of 0.4 μm, the capacitance increases by 0.55 pF. The final parameters of the capacitive converter are given in Table 2.

#### 2.2.2. Electrostatic Vibration Drive

The electrostatic vibration drive is the main source for the excitation of primary vibrations. Its task is to convert electrical vibrations into mechanical ones. The main source of electrostatic excitation is the electric field of the capacitor based on the force of attraction between two parallel plates with opposite charges. The comb counter-pin actuator is shown in Figure 8.

The actuator consists of two combs: the movable rotor and the fixed stator. The tangential force of the comb counter-pin actuator $F_{x}$ is calculated using Equation (Equation 3):(3)$$F_{x}=-\frac{\epsilon_{0}Z_{0}}{y_{0}}NV^{2}$$
where $Z_{0}$ is the thickness of the instrument layer; $Y_{0}$ is the gap between the combs; $X_{0}$ is the overlap area of the fingers of the electrostatic actuator with a variable gap; *N* is the number of pairs of electrostatic fingers; and *V* is the applied voltage on the fingers.

The electrostatic drive was chosen with the greatest possible effort. The technological gap on the edge of the etching technology is 2 μm. The number of combs was selected in such a way as to fit into their dimensions. The parameters of the developed drive are given in Table 3.

The comb counter-pin actuator provides a large amplitude of operation and has a linear characteristic that allows for constant electrostatic force over the entire range of operation. The dependence of the tangential drive force on the voltage on the fingers is shown in Figure 9.

As can be seen in the graph in Figure 9, the force of the electrostatic drive is very small, and within the possible limits, ASIC voltages from 0 V to 8 V correspond to a range from 0 μN to 1.6 μN. The most preferable will be a voltage of about 2 V, at which the force is 0.1 μN.

The comb counter-pin vibration drive or plane-parallel electrostatic actuator has become the most widespread device for the excitation of primary oscillations of MEMS gyroscopes. Another important property of the counter-pin vibration drive is the absence of the effect of electrostatic negative stiffness, since the force function does not depend on displacement. The comb counter-pin vibration drive is shown in Figure 10. The parameters of the developed vibration drive are given in Table 4.

The normal force of a comb counter-pin vibration drive is determined using Equation (Equation 4):(4)$$F_{y}=-\frac{1}{2}{\frac{\epsilon_{0}Z_{0}X_{0}}{\left(y_{0}+y\right)}}^{2}NV^{2}$$

The electrostatic plane-parallel actuator is the main tool for frequency tuning, since the function $F_{y}$ is a nonlinear displacement function and depends on Y. This means that the actuator has the effect of negative electrostatic stiffness, which is determined using Equations (Equation 5) and (Equation 6):(5)$$\frac{1}{2}k_{e}y_{0}^{2}=\frac{1}{2}C_{0}V_{\mathrm{DC}}^{2}$$
(6)$$k_{e}=\frac{C_{0}V_{\mathrm{DC}}^{2}}{y_{0}^{2}}$$

The value of the effective stiffness is determined by Equation (Equation 7):(7)$$k_{\mathrm{eff}}=k_{\mathrm{DM}}-k_{e}$$
where $k_{\mathrm{DM}}$ is the stiffness coefficient in the DM, which is determined using FEM.

Hence, the SM frequency after electrostatic adjustment is determined by Equation (Equation 8):(8)$$f=\frac{1}{2\pi }\sqrt{\frac{k_{e}}{m_{\mathrm{eq}}}}\approx f_{0}-\frac{C_{0}V_{\mathrm{DC}}^{2}}{4\pi m_{\mathrm{eq}}y_{0}^{2}}$$

The constant component of the electrostatic drive always reduces the resonant frequency with an increase in the DC bias on the electrodes. The amount of displacement depends on the initial capacitance and the constant component of the voltage at the electrodes. In this regard, the natural frequency in the SM should have a higher resonant frequency compared to in the DM. The dependence of the electrostatic stiffness on the applied voltage of the comb counter-pin vibration drive is shown in Figure 11.

The capabilities of the Si-Ware ASIC generator allow the voltage to be controlled up to 8 V. Thus, the electrostatic stiffness varies from 0 N/m to 14 N/m.

### 2.3. The Mechanical Part

#### 2.3.1. Elastic Suspension Elements

Elastic suspension elements are the basis of the elastic suspension of the developed MEMS gyroscope design. The operation of the entire elastic suspension depends on the behavior of its individual elements, which should be given special attention, since the accuracy of the calculations and the behavior of the entire elastic suspension depend on the operation and application of various loads. In addition, if possible, various elements should be included in the design of the elastic suspension that facilitate the design and simplify further work on the development of photomasks, analytical and numerical calculations, etc.

In the presented suspension scheme, so-called U-shaped beams are used, the schematic view of which is shown in Figure 12. The U-shaped beam consists of two flat beams and connecting elements connected through a rigid bridge.

The connecting elements of the beam make it possible to facilitate the process of developing the design of the SE and carry out calculations using the FEM method. The geometry of the elastic beam has an asymmetry due to the presence of connecting elements. The **t1–t3** parameters are related to the provision of technological gaps in the manufacture of the SE and make it possible to facilitate the process of developing the design of the SE using computer-aided design (CAD) tools. The parameter **t4** is related to the calculated working amplitude of the MEMS structure: the greater the amplitude, the greater the distance needed between the elastic beams. For more uniform etching and the removal of the asymmetry of elastic elements, it is desirable to make the parameters **t4** and **t2** equal.

The analytical method for calculating the deformations and stiffness of elastic beams is based on the Euler–Bernoulli beam theory. This theory is a special case of the Timoshenko elastic beam theory and is preferred in view of small deformations of the elastic beam, small thickness and the absence of shear deformation [7].

As can be seen in Figure 13, in the suspension of the developed MEMS gyroscope in the operating mode, elastic beams work mainly by bending, since the elastic suspension of the SE has two main degrees of freedom.

This type of deformation corresponds to the deformation of a flat elastic beam with a fixed end; unlike the deformation of an elastic beam with a free end, it always remains parallel to the end with a rigid seal. The limitations of the elastic suspension exclude the angle of rotation of the section φ. The U-shaped elastic beam essentially consists of two flat elastic beams connected by a bridge.

The stiffness coefficient in the direction of force application *F* for an elastic beam with a fixed end is calculated analytically according to the classical theory of bending of Euler–Bernoulli beams using Equation (Equation 9):(9)$$k=E\frac{hb^{3}}{l^{3}}$$
where *h* is the height of the working layer, *b* is the thickness of the elastic beam and *l* is the length of the elastic beam.

In a case in which the elastic elements are connected in parallel, Equation (Equation 10) is valid:(10)$$\frac{1}{k_{\mathrm{total}}}=\frac{1}{k_{1}}+\frac{1}{k_{2}}$$

Thus, the stiffness coefficient for a U-shaped elastic beam with equal arms can be determined using Equation (Equation 11):(11)$$k=E\frac{hb^{3}}{2l^{3}}$$
where *E* is Young’s modulus for silicon, equal to 131 GPa; *l* is the length of the beam.

Analytical equations are used to approximate the stiffness of the elastic suspension and allow the parameters of elastic elements to be selected at the initial stages of designing the SE. For more accurate calculations of both individual nodes and the entire MEMS device, FEM calculations were used. For comparison, we conducted a static analysis of the U-shaped elastic beam using the FEM method and compared it with the analytical calculations. The geometric parameters of the elastic beam used are shown in Figure 14.

Equation (Equation 11) is valid for elastic beams with equal arms; for a symmetrical elastic beam, it is necessary to determine the analytical stiffness of each elastic beam, followed by summation and comparison with the FEM calculations. Table 5 shows the geometric and design parameters of the stiffness of elastic elements and the U-shaped elastic beam as a whole.

The total stiffness of the U-shaped elastic beam when calculated analytically is 22.48 N/m.

For verification, a three-dimensional model of an elastic beam was constructed, and the calculation was performed using the FEM method, taking into account boundary conditions. One end of the elastic beam is rigidly fixed, and a contact restriction is imposed on the second end to move along the plane (excluding the angle of rotation of the section). Hence, according to Hooke’s law ($F_{x}=k\Delta x$), with an elastic beam stiffness of 22.48 N/m, when applying a force of 224.8 μN, the beam displacement should be 10 μm. As a result of FEM, as shown in Figure 15, the displacement of the elastic beam was 10.342 μm.

There are other ways to verify analytical calculations based on ANSYS MultiPhysics CAD tools, for example, by evaluating the reaction of the support when exposed to the deflection of an elastic beam from the zero position, which can be calculated by setting a deviation of 1 μm. As a result of the test, the reaction of the support was 21.736 uN, and the stiffness was 21.736 N/m. At the same time, the error was 3.42%, which is quite natural, since the analytical calculations do not take into account the geometry of the coupling of elastic beams, the place of inflection or the concentration of mechanical stresses.

#### 2.3.2. Selection Criteria for Geometric Parameters of Elastic Elements

When choosing the SE of the MEMS gyroscope, a question may arise when choosing the required parameters of the elastic suspension elements. The first task in the development is to determine the technical and technological limitations in the selection. When manufacturing the MEMS gyroscope by using bulk micromachining technology (SOI technology), they can be:-The operating mode and type of deformation (in-plane or out-of-plane or combined motion by the IM);-The height of the instrument layer (limits the height of the elastic beam);-The minimum and recommended elastic beam width (from the minimum desired value);-The maximum permissible size of the SE of the MEMS gyroscope (limits the length of the elastic beam);-The required stiffness and the amplitude of motion (determines the type of elastic element);-The temperature mode.

Let us consider several options for the implementation of an elastic U-shaped beam. So, we have a technological limitation on the height of the working layer of 50 μm and a technological limitation on the minimum structural element of 3 μm. The rational value of the length of the elastic beam will range from 150 μm, which is 3 times the height of the instrument layer, up to 400 μm, i.e., 8 times the height of the instrument layer. Figure 16 shows a graph of the dependence of the stiffness of an elastic beam on its length.

When choosing the parameters of an elastic beam, it is worth focusing on the following criteria:1.Linear stiffness should be ensured over the entire operating range; i.e., the stiffness of the element must be the same for small deformations and at maximum operating amplitude.2.If the 1st criterion is satisfied by several options, preference should be given to an element with a larger thickness, since, in this way:(a)Technological defects (slopes, etching) have less influence, thereby increasing the yield of suitable products;(b)Higher temperature stability is provided, since the length of the element is shorter, and its elongation under the influence of temperature is also shorter;(c)It is possible to reduce the size of the SE. This opens up more room for the layout of the SE.

#### 2.3.3. Elastic Suspension

The schematic view of the developed elastic suspension of the SE is shown in Figure 17.

The elastic suspension consists of:-The main IM−$M_{1}$;-The mass of the frame of the electrostatic drive−$M_{2}$;-The mass of the frame of the capacitive converter−$M_{3}$;-The elastic elements−$k_{1}$–$k_{4}$.

For the construction, a previously calculated elastic element with a stiffness equal to 21.736 N/m was used, and the remaining suspension parameters are given in Table 6.

When oscillations are excited in the DM, IMs $M_{1}$ and $M_{2}$ are driven into motion; the equivalent mass of the DM is calculated using Equation (Equation 12):(12)$$M_{\mathrm{DM}}=M_{1}+2M_{2}+8M_{k}$$

Thus, the equivalent mass of the DM is 1.913 × 10^−7^ kg.

Under the influence of the Coriolis force, masses $M_{1}$ and $M_{3}$ and elastic element $k_{4}$ are set in motion, and elements $k_{1}$ and $k_{3}$ are deformed. The equivalent mass of the SM is calculated using Equation (Equation 13):(13)$$M_{\mathrm{SM}}=M_{1}+2M_{3}+8M_{k}$$

Thus, the equivalent mass of the SM is 1.751 × 10^−7^ kg.

As can be seen, the equivalent mass of the SM is greater than the equivalent mass of the DM. When elastic elements are connected in parallel, the ratio is valid: $k_{\mathrm{total}}=k_{1}+k_{2}+k_{3}+\ldots +k_{n}$.

When calculating the total stiffness of the suspensions in the modes of motion, $k_{\mathrm{DM}}$ and sensitivity $k_{\mathrm{SM}}$ are equal to 173.89 N/m; when analyzing the FEM, the reaction of the support with a deflection of 1 μm is 167.4 uN, and the stiffness is 167.4 N/m.

#### 2.3.4. Modal Analysis

Modal analysis, along with static analysis, is one of the main tools of FEM used in the development of MEMS devices. This tool allows the determination of the mode forms and natural frequencies of vibrations, i.e., the qualitative and quantitative types of analysis of a micromechanical design. The first mode of oscillation characterizes the direction of motion with minimal stiffness. There are two such modes for the SE of the LL-type MEMS gyroscope developed: DM and SM. The first oscillation mode of the developed SE is shown in Figure 18a. The natural frequency according to the calculation results is 5655 Hz.

The first mode of oscillation is directed along the axis of motion. To excite it, it is necessary to supply a control voltage. According to the formula for calculating the natural frequency, $\omega_{0}=\sqrt{\frac{k}{m}}$. The second form of oscillation is shown in Figure 18b. The natural frequency is 5710 Hz. The second oscillation mode is directed along the sensitivity axis and serves to extract the sensor signal from the sensor. Additionally, this axis can be used to measure the acceleration of the MEMS gyroscope–accelerometer.

The third mode of oscillation is parasitic and is associated with the rotation of the IM of the MEMS gyroscope design being developed relative to the attachment points of elastic suspensions. The frequency of the third mode is 7448 Hz and is quite different from the frequency of the first and second modes. The third form of oscillation is shown in Figure 18c.

The fourth and subsequent modes are associated with the vibrations of individual structural elements of the gyroscope being developed by MEMS and are at the frequency of ultrasonic vibrations.

### 2.4. Modeling of the SE

#### 2.4.1. Methods of Studying the Dynamic Characteristics

The majority of SEs of MEMS gyroscopes are vibrational resonant structures. The design of the SE can be considered a combination of two oscillatory systems with one degree of freedom: the first system characterizes the DM, and the second one characterizes the SM. Thus, understanding the dynamics and frequency characteristics of a resonant system with one degree of freedom is crucial in the design of the SE.

The basic dynamic properties of a resonant system whose vibrations are caused by an external force can be considered a model of a freely vibrating oscillator. Such a model best characterizes the majority of SEs, since forces are constantly acting on them; in the DM, there is a force that excites primary vibrations−the force of the electrostatic actuator; in the SM, it is the Coriolis force.

The equation of motion of an oscillator with viscous damping is described by a second-order differential Equation (Equation 14):(14)$$m\ddot{x}+c\dot{x}+kx=F\left(t\right)$$
where *m*, *c* and *k* are, respectively, the mass, the damping and the mechanical stiffness.

The natural frequency of undamped oscillations is determined by Equation (Equation 15):(15)$$\omega_{0}=\sqrt{\frac{k}{m}}$$

The frequency response of the second-order equation is characterized by the damping coefficient ξ, which expresses the level of damping relative to the critical damping and is determined by Equation (Equation 16):(16)$$\xi =\frac{c}{c_{k}}=\frac{c}{2\sqrt{\mathrm{km}}}=\frac{c}{2m\omega_{0}}$$

Taking into account Equations (Equation 15) and (Equation 16), Equation (Equation 14) takes the form of Equations (Equation 17) and (Equation 18):(17)$$\ddot{x}+2\xi \omega_{0}\dot{x}+\omega_{0}^{2}x=\frac{F(t)}{m}$$
(18)$$\frac{X(S)}{F(S)}=\frac{1/m}{s^{2}+2\xi \omega_{0}S+\omega_{0}^{2}}$$

Another important parameter that determines the dynamic characteristics of the microsystem is the *Q*-factor. The exact expression for the *Q*-factor is the ratio of stored energy *E* to energy losses over a period of time Δ*E* and is determined using Equation (Equation 19):(19)$$Q=2\pi \frac{E}{-\Delta E}$$

This means that the higher the *Q*-factor, the less energy the system loses for each period and the slower the oscillations fade.

Following this definition, the *Q*-factor can be expressed in terms of the natural frequency $\omega_{n}$ and the logarithmic decrement of attenuation δ or the period *T*. In this case, the *Q*-factor is determined by Equation (Equation 20):(20)$$Q=\frac{\omega_{n}}{2\delta }=\frac{2\pi }{1-e^{-2T\delta }}$$

#### 2.4.2. Investigation of Amplitude–Frequency Characteristics of FEM

Harmonic analysis is the next step in the simulation of the SE of the MEMS gyroscope and is carried out to solve the equations of motion in the case of steady-state oscillatory processes. Harmonic analysis is carried out using the method of mode superposition, in which natural frequencies and waveforms are used to analyze steady-state forced harmonic oscillations; i.e., the results of modal analysis are applied. Harmonic analysis was performed using the method of mode superposition with ANSYS MultiPhysics CAD tools.

Since the shape of the resonance curve is nonlinear, the Lorentz and Gauss functions are best suited for postprocessing the results of experimental measurements and approximating the values. The peculiarity of postprocessing is that the bandwidth of the Lorentz function and the Gauss function is calculated not by the half-power level, but by the half-amplitude level, i.e., not by the 0.707 level, but by the 0.5 level from the top of the resonant peak. To conduct a full-fledged analysis, it is necessary to determine the damping coefficient of the structure. Damping assessment without experimental data can be carried out analytically.

#### 2.4.3. Analytical Model of the Damping

One of the important characteristics in assessing damping is the Reynolds number $R_{e}$, which characterizes the relationship between inertia and friction forces and, thus, allows us to draw conclusions about the characteristics of flow processes. The Reynolds number can be determined using Equation (Equation 21):(21)$$R_{e}=\frac{\rho \nu d}{\eta }$$
where ρ is the density of the medium; ν is the characteristic velocity; *d* is the hydraulic diameter; and η is the dynamic viscosity of the medium.

Microstructures are characterized by small dimensions and small hydraulic diameters when the Reynolds number becomes very small. In this case, the influence of viscous friction forces increases, and all flows have a laminar character.

Another distinctive feature of the microstructure is operation at low pressure, at which it is necessary to take into account the molecular structure of the liquid. The degree of sparsity of the flow is characterized by the Knudsen number $K_{n}$ and is determined using Equation (Equation 22):(22)$$K_{n}=\frac{\lambda }{L}$$
where λ is the average free path of molecules in the gas; *L* is the characteristic size of the flow.

The free path length of gas molecules is inversely proportional to pressure and is determined by Equation (Equation 23):(23)$$\lambda =\frac{p_{0}}{p}\lambda_{0}$$

Lateral damping depends on the conditions of gas adhesion to the moving surface and the viscosity of the liquid and is determined by the absolute coefficient of lateral damping $c_{l}$, which can be found using Equation (Equation 24):(24)$$c_{l}=\mu \frac{S}{d}$$

Another damping mechanism shown is related to the compression of the gas between two plates moving toward each other. Due to vertical movement, a pressure drop occurs in the gap between the plates, and Poiseuille flow develops. In this case, the damping coefficient is a function of frequency and is determined using Equation (Equation 25):(25)$$\omega =\frac{12\mu \omega l^{2}}{P_{0}h_{0}^{2}}$$

To calculate the damping coefficient for the design of the SE, an analysis of the components was carried out for the damping introduced by each element into the overall design.

The IM has dimensions of 1.2 mm × 1.2 mm and occupies a large area of the MEMS gyroscope: 1.395×10^−6^
$m^{2}$. A schematic view of the IM is shown in Figure 19. The distance between the IM and the substrate is 20 μ
m.

The electrostatic actuator has a surface area of 1.265×10^−7^
$m^{2}$ with a distance from the substrate of 20 μm, and the area between the combs is 3×10^−7^
$m^{2}$ with a distance between fingers of 2 μm, which is 10 times less than that between the base and IM. A schematic view of the electrostatic actuator is shown in Figure 20. Two electrostatic actuators contribute to the DM.

The capacitive converter has the largest overlap area with the type of compression damping, in addition to lateral damping. The overlap area is 5×10^−7^
$m^{2}$ with a gap of 2 μm, and the same area can be calculated with a gap of 5 μm, since each comb has a working area on both sides. A schematic view of the electrostatic actuator is shown in Figure 21.

It is also worth noting that two capacitive converters need to be taken into account in the SM. As a result of the analytical evaluation, the damping coefficient $\xi_{\mathrm{DM}}$ for the DM is approximately $\xi_{\mathrm{DM}}=5\times 10$^−4^. In the DM, the damping coefficient of $\xi_{\mathrm{SM}}$ is approximately 10 times higher due to the damping of the compression of capacitive converters and is on the order of $\xi_{\mathrm{SM}}=1.5\times 10$^−3^.

#### 2.4.4. Amplitude–Frequency Characteristic on the Axis of Motion

In order to construct the frequency response of the SE, a harmonic analysis of FEM was performed with the following boundary conditions:(a)Support of oscillation modes: 2;(b)The force of the electrostatic drive is 0.3 uN, which is approximately ±4 V;(c)The damping coefficient is $\xi_{\mathrm{DM}}=5\times 10$^−4^.

To determine the frequency response, the “Frequency Response” tool was used, taken from the values of the average deformation of the surface IM. The results obtained using ANSYS MultiPhysics CAD tools are shown in Figure 22.

A series of calculations with different damping coefficients was carried out to verify the adequacy of the mathematical model. Figure 23 shows a series of graphs with different damping coefficients.

After the Lorentz approximation of the mathematical experimental data, the graph shown in Figure 24 was obtained.

#### 2.4.5. Amplitude–Frequency Characteristics on the Sensitivity Axis

To construct the amplitude–frequency response along the sensitivity axis, the magnitude of the Coriolis force acting on the IM in the SM was estimated according to Equation (Equation 26):(26)$$F_{k}=2Mx\omega_{x}\Omega$$
where *x* is the amplitude of the SE along the X-axis; *M* is the IM of the SE; $\omega_{x}$ is the natural frequency of vibrations of the SE along the X-axis and equal to 4747 Hz (modal analysis results); and Ω is the angular velocity in the range 0–5 rad/s.

The graph of the Coriolis force’s dependence on the angular velocity is shown in Figure 25.

## 3. Manufacturing of the SE

The SE of the MEMS gyroscope being developed is formed from 3 plates: (**1**)−the base; (**2**)−the instrument layer; and (**3**)−the package. For the prototype, the package was not manufactured due to the necessary research, and there is no eutectic layer to save development. The layered structure of the prototype is shown in Figure 26, where (**1**) is a glass base with a recess for movable structures; (**2**) is a bottom layer with the developed topology of the SE; (**3**) is a metallization layer for soldering a thin wire into the package. The technological route for manufacturing the developed SE in accordance with bulk micromachining technology (SOI technology) is schematically shown in Figure 27.

All design standards represent restrictions on the geometric dimensions of topological elements. In particular, these are the minimum and maximum allowable dimensions, the minimum allowable gaps between elements, area restrictions, etc. The topology of technologically predefined elements is fixed and cannot be changed by the developer. The process parameters are presented in Table 7.

The ultimate goal of developing the design of the MEMS gyroscope is to create a topology, that is, the geometry of contours in all topological layers. As part of this technological process, all the layers listed in Table 8 must be formed.

Based on this topology, a set of photomasks was made, which were then used at the stage of photolithography. Sets of photomasks were developed in GDSII format in the KLayout software environment.

Before the production of the photomasks, the final development of the SE was carried out, when the topology of the SE was inscribed in the dimensions of the crystal and the metallization wiring was applied.

The final version of the developed SE of the MEMS gyroscope is shown in Figure 28.

## 4. Experimental Studies of a Prototype MEMS Gyroscope

The topology of the SE of the MEMS gyroscope was studied using the Nova Nanolab 600 SEM with an optical measuring head at the Research Laboratory of Functional Nanomaterials Technologies of the Institute of Nanotechnology, Electronics and and Equipment Engineering, Southern Federal University, Taganrog, Russia.

As a result of the measurements of the samples, as shown in Figure 29, the dimensions of the elastic elements of the suspensions, the dimensions of the decoupling frame, the dimensions of the comb structures, the gaps between the combs of the electrostatic drive and the capacitive converter and the gap between the elastic locking elements were determined.

To measure the height of the structure and the amount of etching of the mask, as shown in Figure 30, several SEs had to be destroyed.

The measurements of the geometric dimensions of the capacitive converter are shown in Figure 31.

When examining samples, the measurement accuracy was affected by the lack of special equipment for fixing the SE, which allows the element to be positioned strictly horizontally, strictly vertically or at a required angle. The difficulty in making measurements with a scanning microscope was the dielectric glass base, which did not allow contact between the substrate and the structure under study. In this case, specialized equipment provided high-quality contact with the SE for the supply of an acceleration voltage. The accuracy of measuring geometric dimensions is determined by the operator’s measurement error and was no more than 5% of the nominal topological dimensions, which indicates the high quality of the manufactured test structures of the developed MEMS gyroscope. The results of topological measurements of the SE are presented in Table 9.

For testing the samples, the SE of the MEMS gyroscope must be pre-packaged and soldered. The ceramic package “QLCC48” was used for this purpose, and similar packages were used for testing [51]. The use of this type of package is due to the equipment being placed in a vacuum chamber, as shown in Figure 32.

In the study of the samples, scanning laser vibrometry technologies were used to study the vibration behavior at a point, and stroboscopic video microscopy was used for the accurate measurement of high-frequency vibrations in the plane. To measure the amplitude–frequency characteristics, a Polytec MSA-500 microsystem analyzer with a vacuum chamber was used.

To measure the free oscillations of the IM of the MEMS gyroscope, it is necessary to deflect the IM from the initial position and then measure the amplitude of the free oscillations. It is possible to remove the IM from the initial position either by applying an external influence—by shock or vibration—or by using electrostatic transducers for most SEs. To do this, it is necessary to use electrostatic structures to excite the oscillations of the system with a sinusoidal voltage in the frequency sweep mode and simultaneously detect the response.

The measured amplitude–frequency characteristics of the prototypes of the MEMS gyroscope are shown in Figure 33 in the DM.

The measured amplitude–frequency characteristics of the prototypes of the MEMS gyroscope in the SM are shown in Figure 34.

In the study of the prototypes, scanning laser vibrometry technologies were used to study the spatial vibration behavior at a point on the prototype, and stroboscopic video microscopy was used for the accurate measurement of high-frequency vibrations in the plane during device tests. When using stroboscopic illumination and digital images, rapid periodic movements of an object can be instantly stopped to capture the position in the area of interest of the test prototype.

In Table 10, the results of an experiment to determine the frequency response of experimental samples of the MEMS gyroscope are presented.

## 5. Conclusions

In this article, on the basis of the developed design methodology, the SE of a MEMS gyroscope was calculated, designed and then manufactured. Structures were made using bulk micromachining technology−silicon-on-insulator (SOI technology)−with a structural layer thickness equal to 50 μm. The manufactured structures were examined using SEM, profilometric studies were carried out, and the amplitude–frequency response was studied at various degrees of vacuuming on a Polytec MSA-500 microsystem analyzer.

The development started from the ASIC used, which influenced the choice of used containers and electrostatic actuators. The mechanical part of the structure was synthesized from an elastic U-beam. Minor differences in the modeling and experimental results can be explained by technological errors. A patent from the Russian Federation has been obtained for this topology of SE of the MEMS gyroscope [52,53]. The results obtained will be used by us as a basis for the design of multi-mass structures with synchronizing elements.

Due to sanctions against Russia, the supply of SW 6111 chips has been suspended. As a result, work began on the development of an ASIC based on Russian 5201TK015 chips [54]. The signal is extracted using a 5201TK015 capacitance–voltage converter developed by the Joint Stock Company “Zelenograd Nanotechnology Center” (Zelenograd, Moscow, Russia). The IC is a capacitance–voltage converter with an analog output and a programmable conversion path. The digital interface SPI allows the programming of basic parameters of the converter, such as gain, reference voltage and correction for nonlinearity. To build temperature-independent systems, a built-in temperature sensor can be used. The circuit has a block forming a reset when the power is turned on and an overload detector.

On the basis of this chip, layouts were made on discrete components on a printed circuit board, as shown in Figure 35.

When using the 5201NR015 circuit, the connection scheme is similar to the SW6111 circuit; however, a primary oscillator circuit will be required, which can be implemented on a microcontroller, for example, manufactured by the Research Institute of Measuring Systems named after Yu.E. Sedakova (Nizhniy Novgorod, Russia) [55].

Bulk micromachining technology (SOI technology) for the manufacture of structures with instrument layer thicknesses of 60 μm and 100 μm has been mastered at several Russian enterprises, and the sensor design will require changes. Additionally, the new sensor design is built on a multi-mass architecture with synchronizing elements. Based on the results of the layout, an analog–digital chip for a new topology of SE of the MEMS gyroscope adapted to the technological capabilities of Russian enterprises will be developed.

## Figures and Tables

**Figure 1 micromachines-14-00895-f001:**
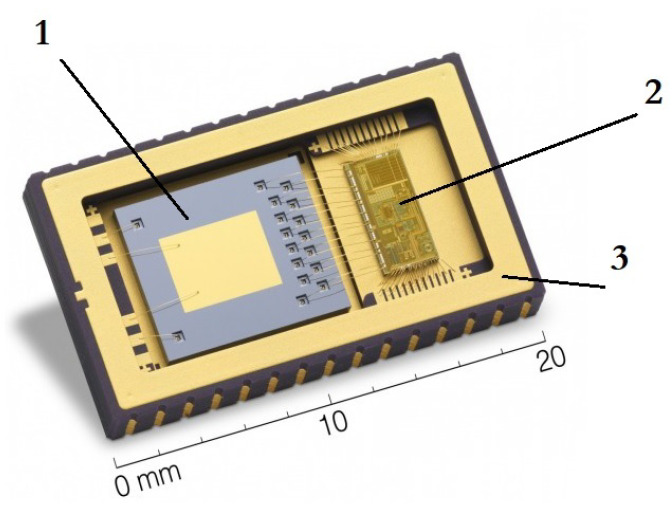
MEMS gyroscope (Tronics Gypro2300LD). (**1**) SE; (**2**) the ASIC; (**3**) package.

**Figure 2 micromachines-14-00895-f002:**
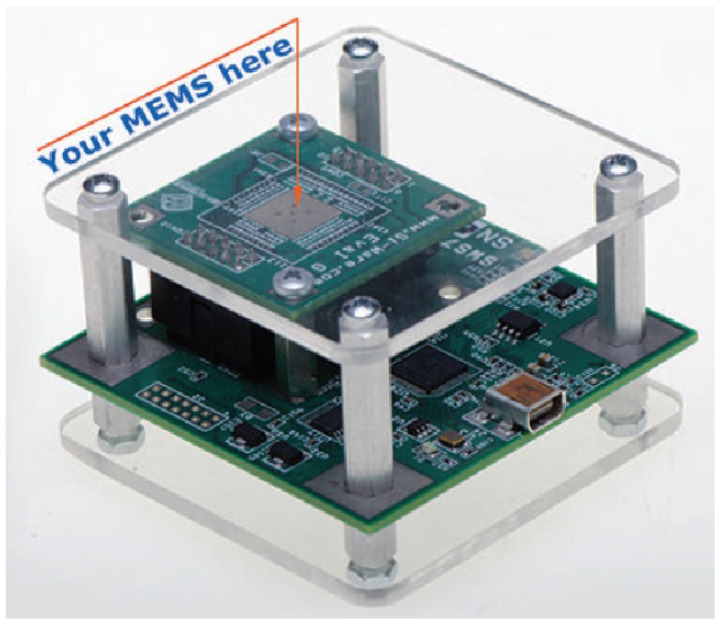
Schematic view of the specialized SWS61111 platform.

**Figure 3 micromachines-14-00895-f003:**
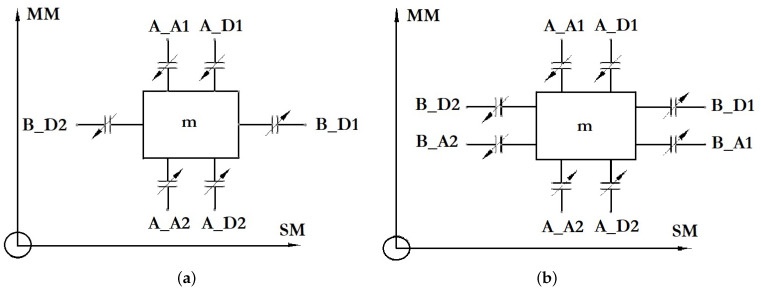
Scheme of the SE of the MEMS gyroscope for ASIC SWS1110: (**a**) open-loop structure; (**b**) closed-loop structure.

**Figure 4 micromachines-14-00895-f004:**
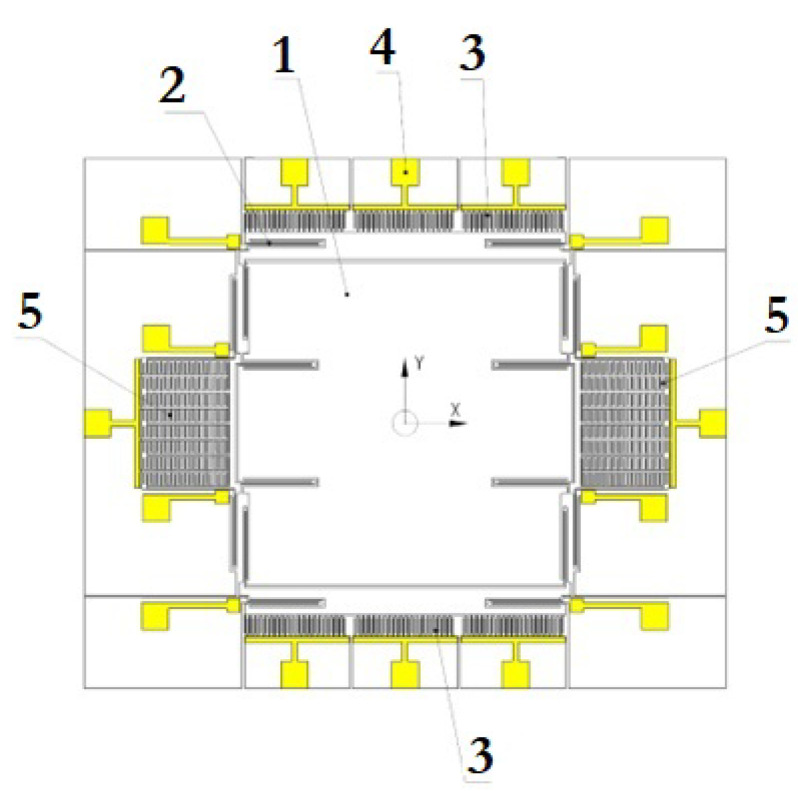
The developed structure: the topology of the SE of the MEMS gyroscope, (**1**) IM, (**2**) U-beam, (**3**) electrostatic drive, (**4**) aluminum contact pads, (**5**) capacitive plates.

**Figure 5 micromachines-14-00895-f005:**
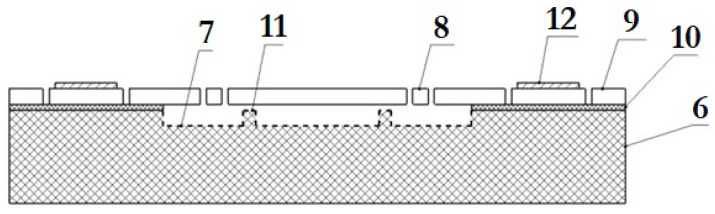
The developed structure: the cross-section of the SE of the MEMS gyroscope. (**6**) substrate of glass, (**7**) cavities, (**8**) elastic elements, (**9**) silicon instrument layer, (**10**) (SiO_2_), (**11**) support, (**12**) metallization layer.

**Figure 6 micromachines-14-00895-f006:**
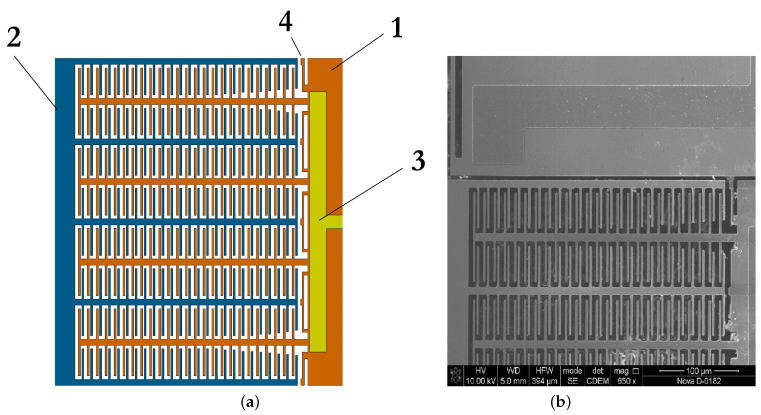
Capacitive displacement converter: (**a**) schematic view, stator (**1**), rotor (**2**), contact pad (**3**), elastic locking elements (**4**); (**b**) SEM image.

**Figure 7 micromachines-14-00895-f007:**
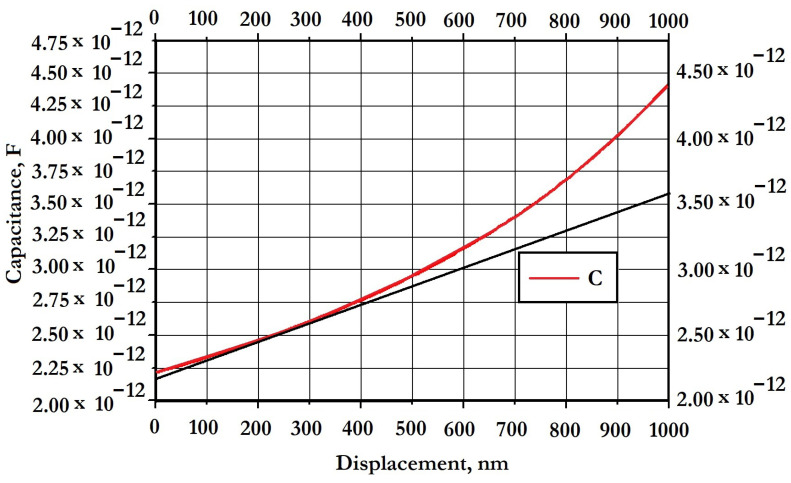
Graph of capacitance changes when the gap between the fingers decreases.

**Figure 8 micromachines-14-00895-f008:**
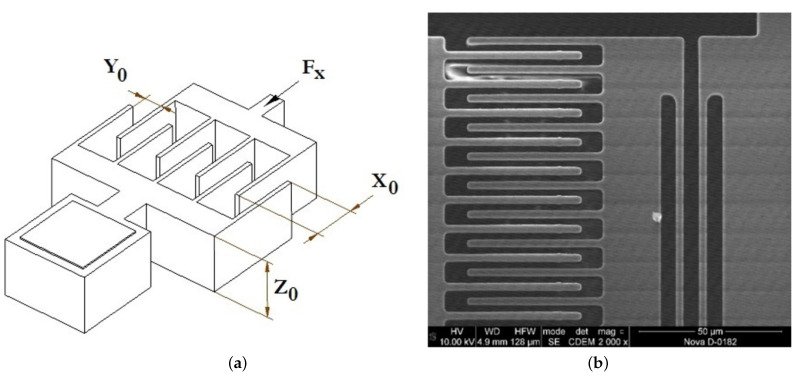
Electrostatic comb counter-pin actuator: (**a**) schematic view; (**b**) SEM image.

**Figure 9 micromachines-14-00895-f009:**
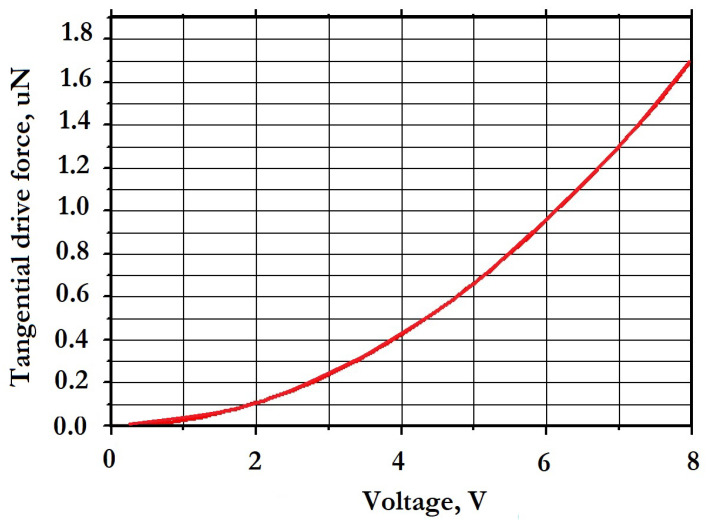
Graph of the dependence of the electrostatic drive force on voltage.

**Figure 10 micromachines-14-00895-f010:**
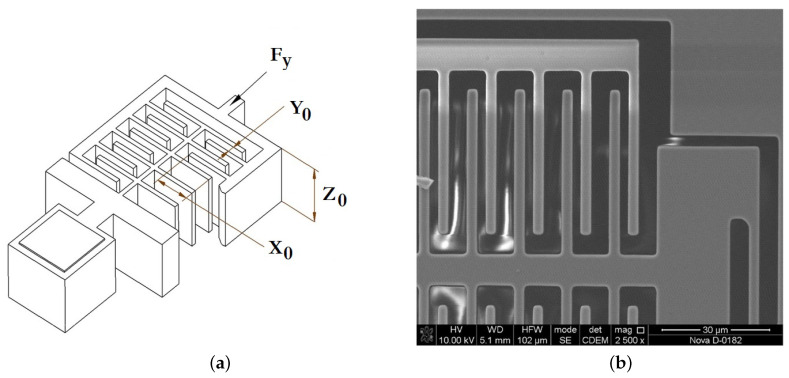
Electrostatic comb counter-pin vibration drive: (**a**) schematic view; (**b**) SEM image.

**Figure 11 micromachines-14-00895-f011:**
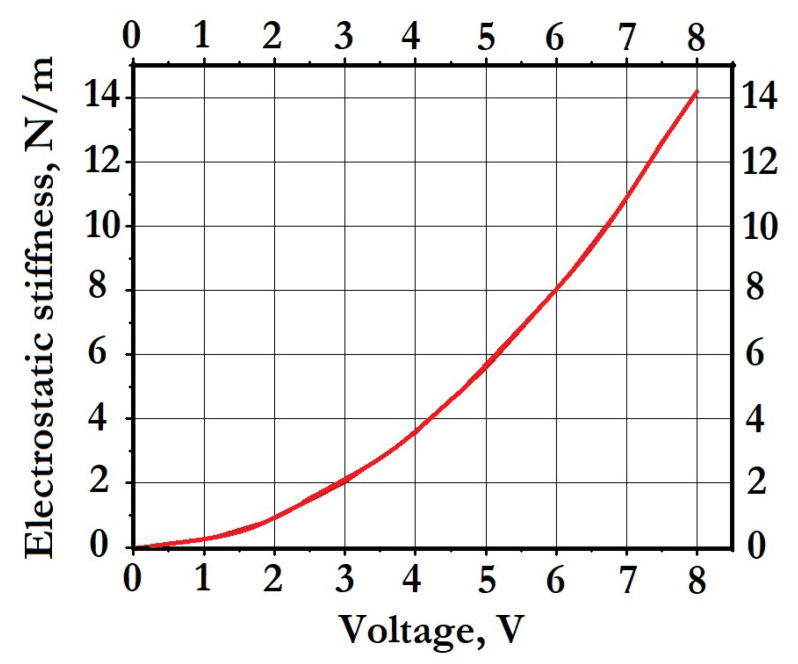
Graph of the dependence of the electrostatic stiffness on the applied voltage.

**Figure 12 micromachines-14-00895-f012:**
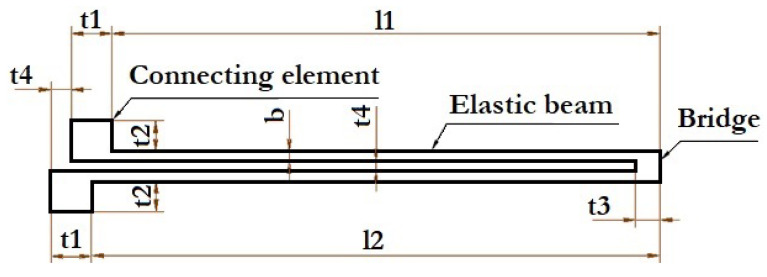
Schematic view of the U-shaped elastic beam used.

**Figure 13 micromachines-14-00895-f013:**
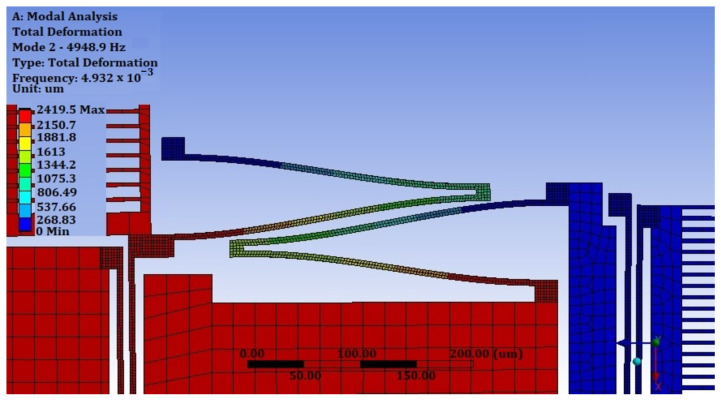
The diagram of deformations of an elastic beam during bending.

**Figure 14 micromachines-14-00895-f014:**
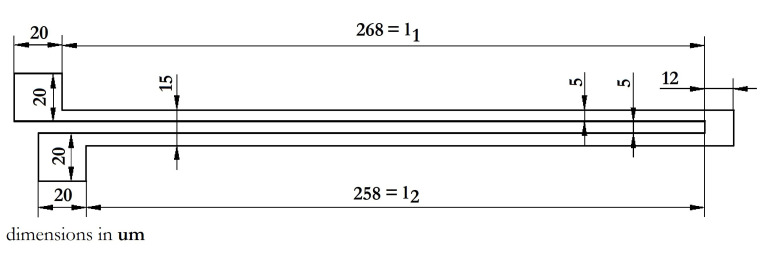
Geometric parameters of a U-shaped elastic beam.

**Figure 15 micromachines-14-00895-f015:**
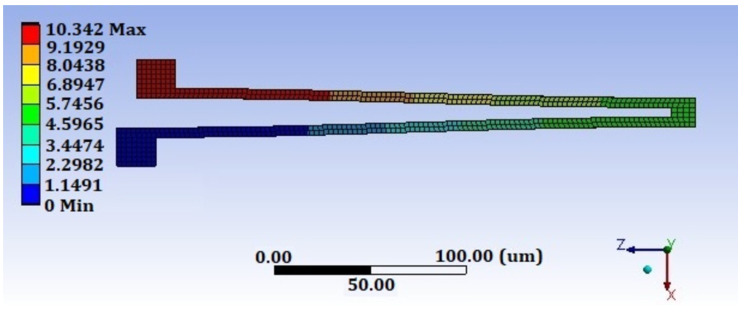
The results of checking the analytical calculations of the elastic beam using FEM.

**Figure 16 micromachines-14-00895-f016:**
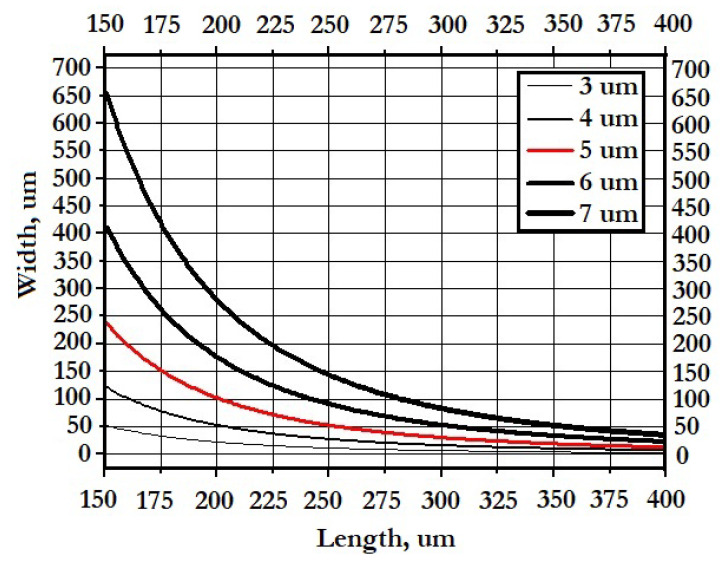
Dependence of the stiffness of the U-shaped elastic beam on the design parameters.

**Figure 17 micromachines-14-00895-f017:**
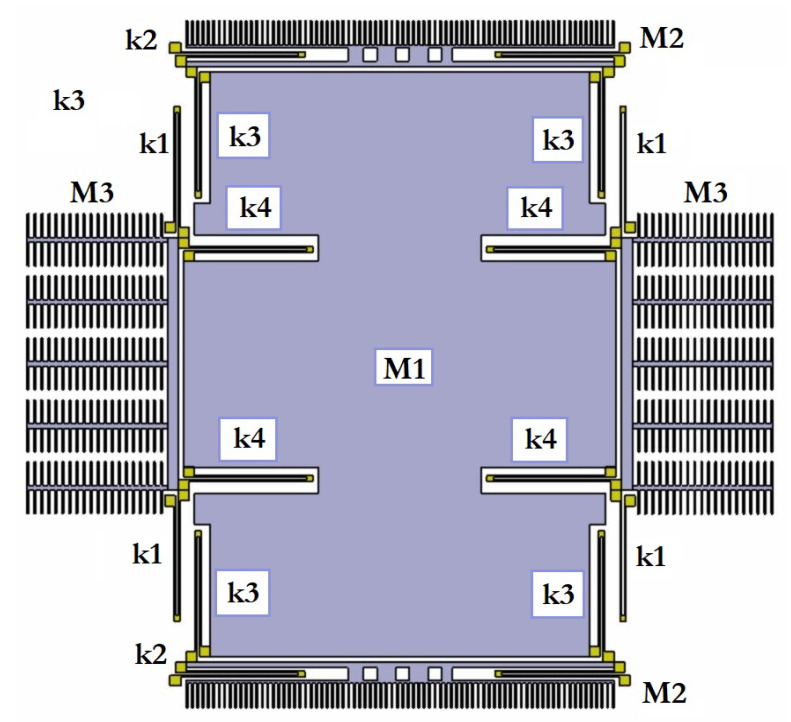
Schematic view of the elastic suspension of the SE of the MEMS gyroscope.

**Figure 18 micromachines-14-00895-f018:**
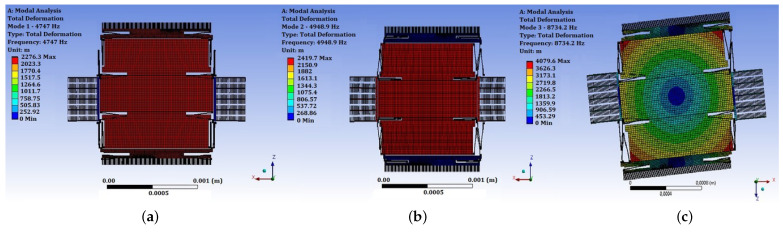
Modes of oscillation of the SE of the MEMS gyroscope: (**a**) the first oscillation mode; (**b**) the second oscillation mode; (**c**) the third oscillation mode.

**Figure 19 micromachines-14-00895-f019:**
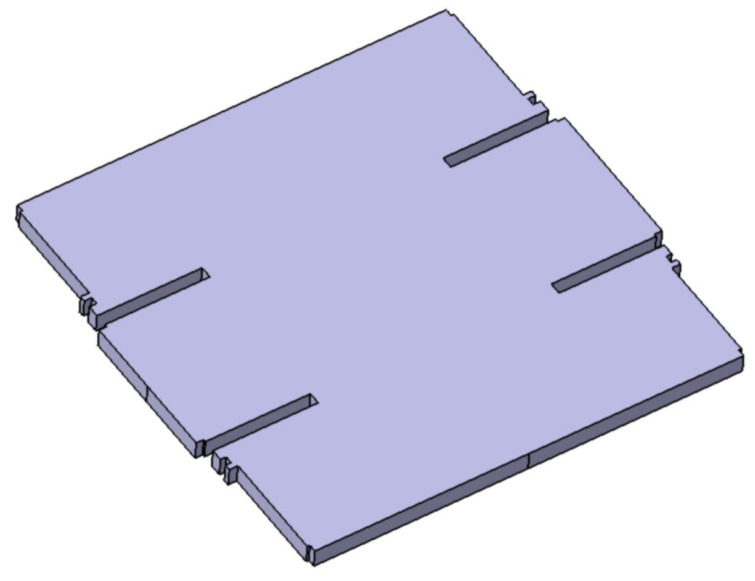
A schematic view of the IM of the SE of the MEMS gyroscope.

**Figure 20 micromachines-14-00895-f020:**
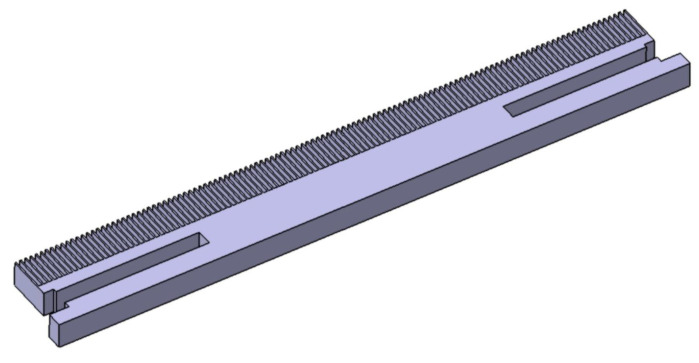
A schematic view of the electrostatic actuator of the SE of the MEMS gyroscope.

**Figure 21 micromachines-14-00895-f021:**
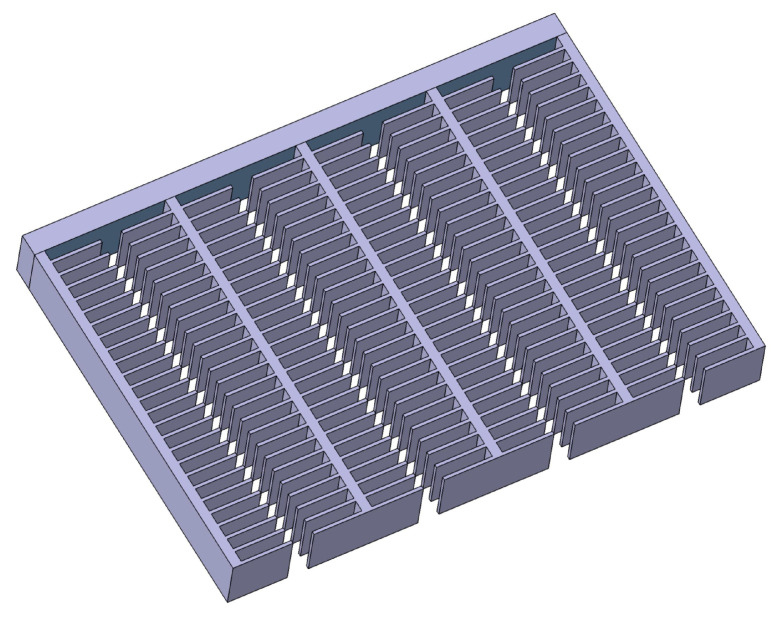
A schematic view of the capacitive converter of the SE of the MEMS gyroscope.

**Figure 22 micromachines-14-00895-f022:**
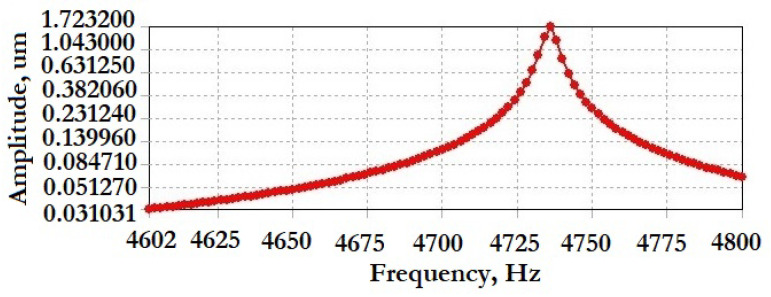
Calculated amplitude–frequency response of the DM.

**Figure 23 micromachines-14-00895-f023:**
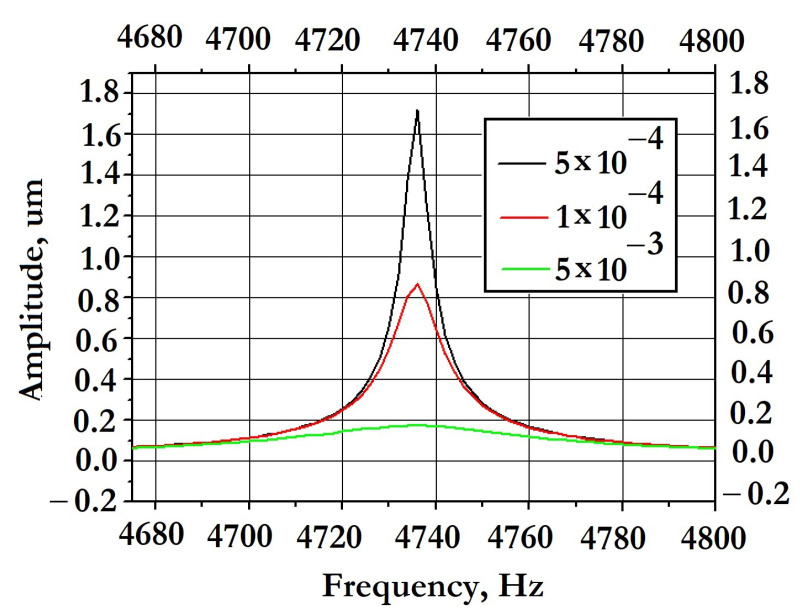
Calculated amplitude–frequency response of the DM with different damping.

**Figure 24 micromachines-14-00895-f024:**
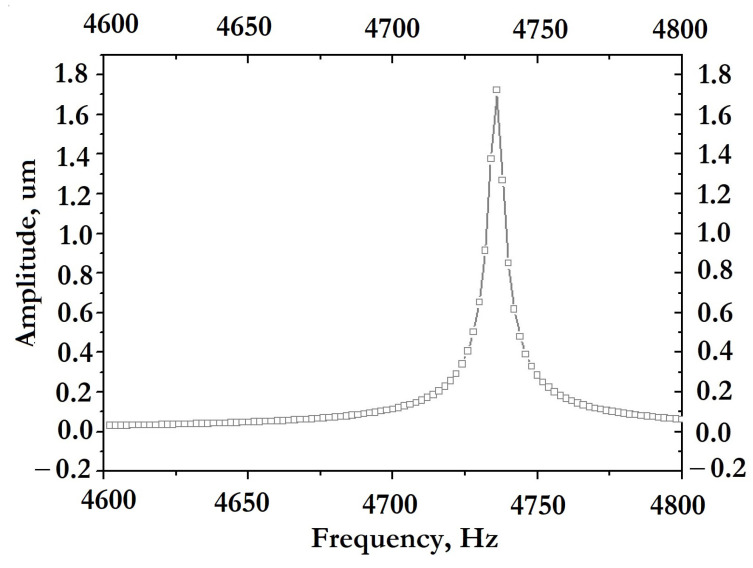
Approximation by the Lorentz method using data from a mathematical experiment on finding the amplitude–frequency response of the SE.

**Figure 25 micromachines-14-00895-f025:**
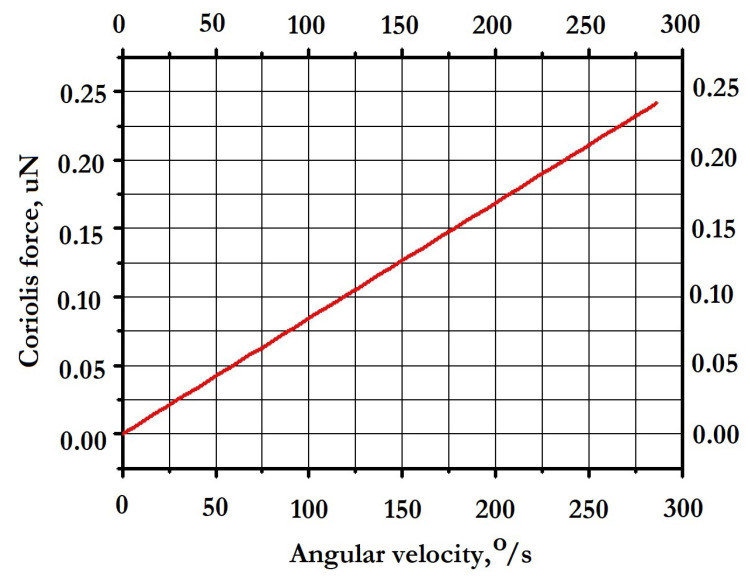
Graph of Coriolis force’s dependence on angular velocity.

**Figure 26 micromachines-14-00895-f026:**
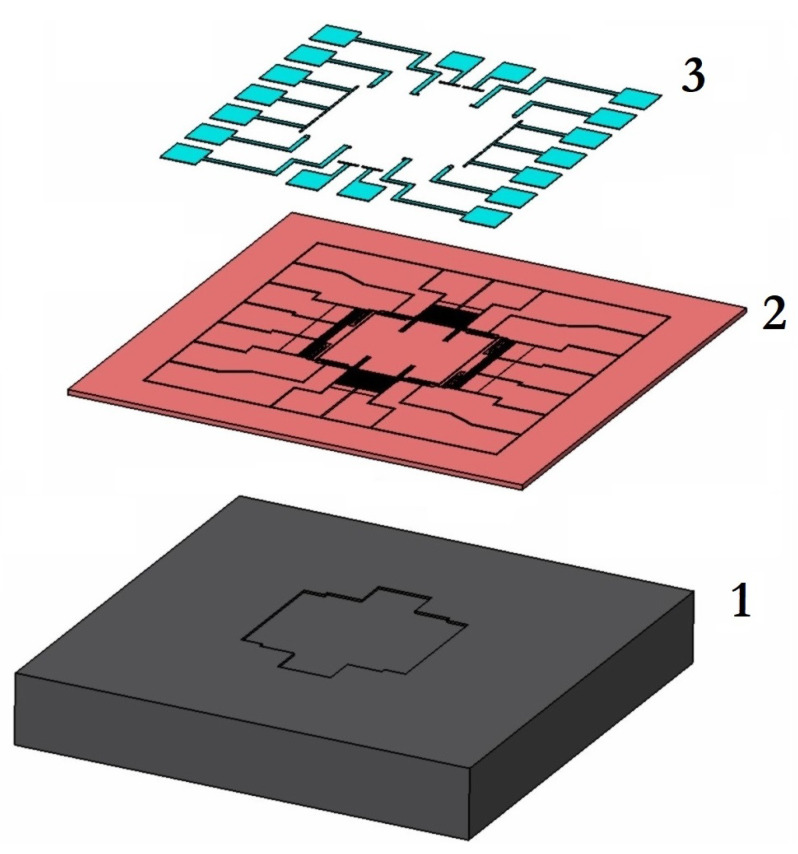
Layered structure of the SE of the MEMS gyroscope. (**1**) a glass base with a recess for movable structures; (**2**) a bottom layer with the developed topology of the SE; (**3**) a metallization layer for soldering a thin wire into the package.

**Figure 27 micromachines-14-00895-f027:**
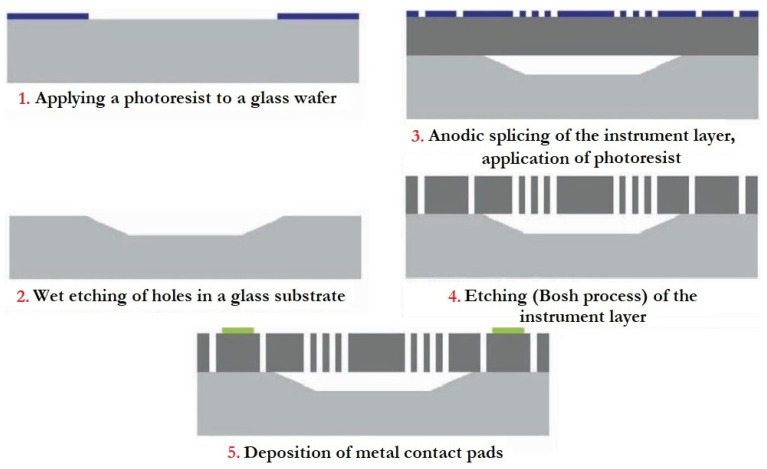
Technological route for the manufacturing of the developed LL-type MEMS gyroscope.

**Figure 28 micromachines-14-00895-f028:**
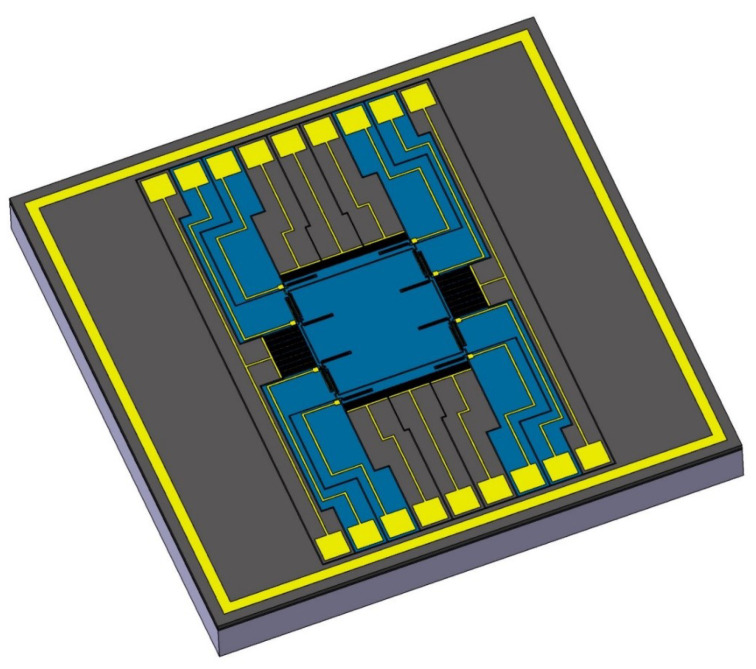
The final model of the SE of the MEMS gyroscope.

**Figure 29 micromachines-14-00895-f029:**
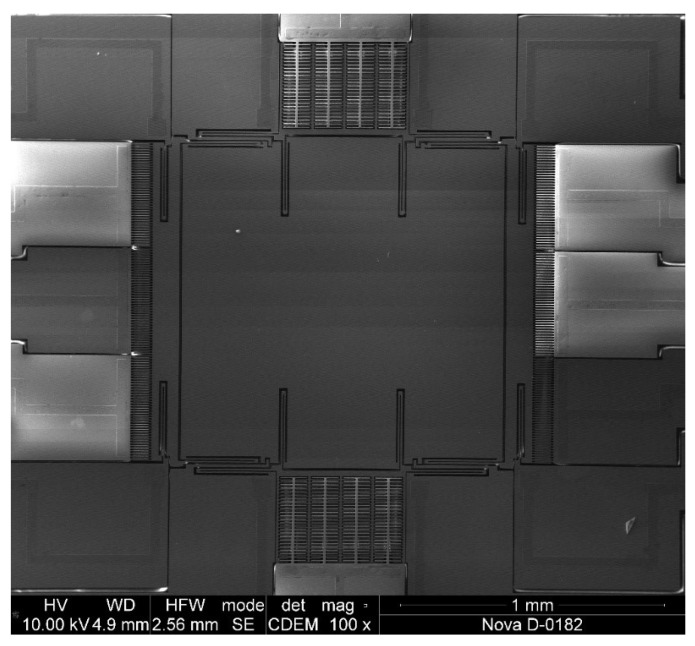
General view of the MEMS gyroscope sample under study using SEM.

**Figure 30 micromachines-14-00895-f030:**
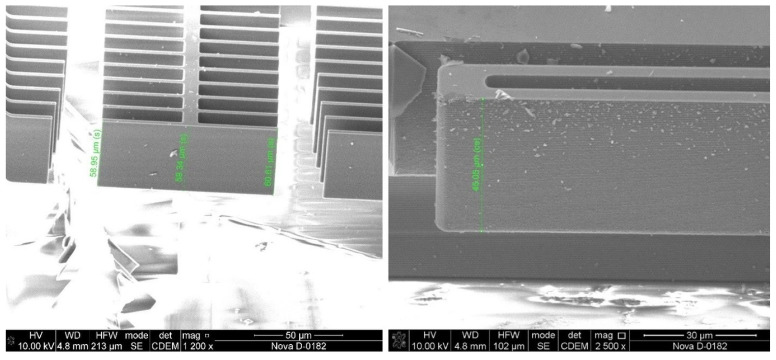
Measuring the height of the instrument layer by SEM.

**Figure 31 micromachines-14-00895-f031:**
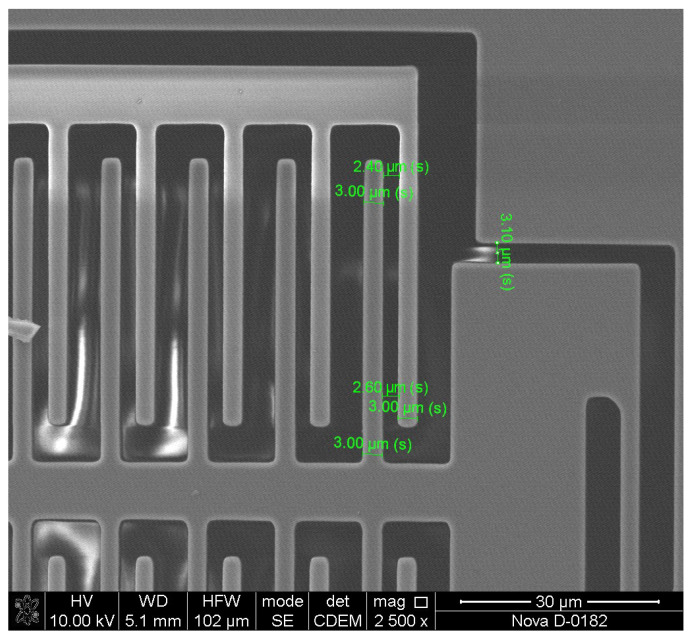
Measurement of the topology of a capacitive converter by SEM.

**Figure 32 micromachines-14-00895-f032:**
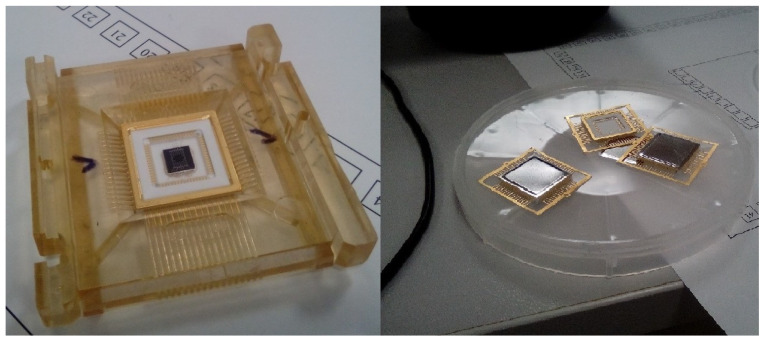
The studied prototype of the SE of the MEMS gyroscope on the measuring equipment.

**Figure 33 micromachines-14-00895-f033:**
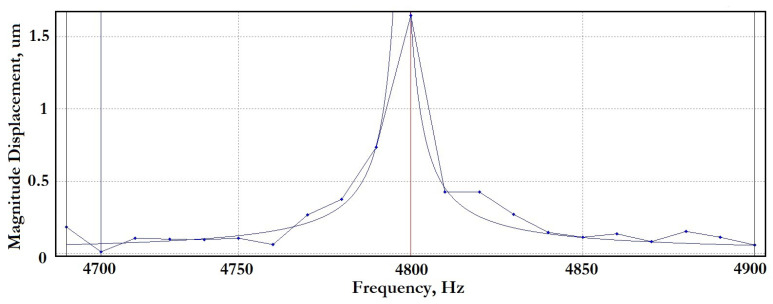
The measured amplitude–frequency response of the MEMS gyroscope in the DM.

**Figure 34 micromachines-14-00895-f034:**
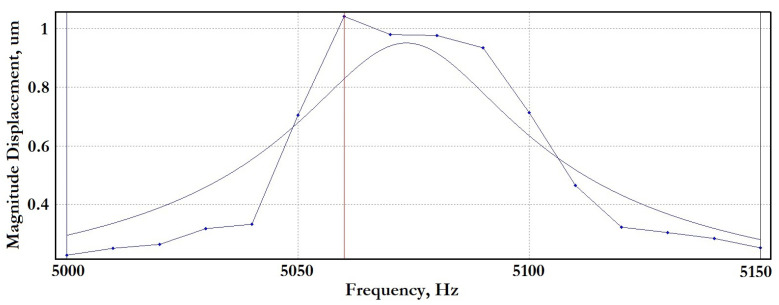
The measured amplitude–frequency response of the MEMS gyroscope in the SM.

**Figure 35 micromachines-14-00895-f035:**
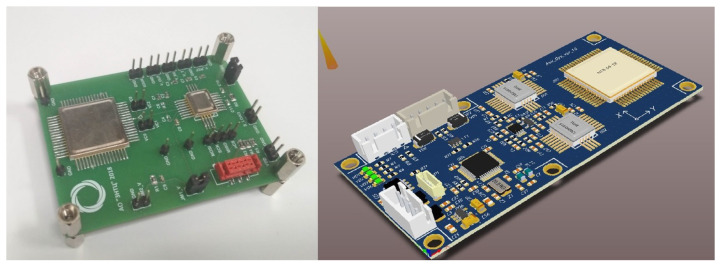
Debugging board based on one and two 5201TK015 chips.

**Table 1 micromachines-14-00895-t001:** Design parameters of the capacitive element of the SE.

Parameter	Value
Total number of comb fingers, $N_{f}$	250
Finger height, $z_{0}$	50 μm
Overlap length, *l*	40 μm
Finger overlap area, *S*	5×10^−7^, $m^{2}$
The size of the air gap between the fingers, $d_{0}$	2 μm
Estimated capacitance	2.2 pF

**Table 2 micromachines-14-00895-t002:** Parameters of the differential capacitance of the SE.

Full differentiation capacitance	4.4 pF
Final differential capacitance (displacement 0.4 μm)	5.5 pF
Sensitivity	2.5 pF/μm

**Table 3 micromachines-14-00895-t003:** Parameters of the comb counter-pin actuator.

Parameter	Value
Comb height, $Z_{0}$	50 μm
Number of pairs of fingers, *N*	120
The length of the overlap of the combs, $X_{0}$	60 μm
The gap between the fingers, $Y_{0}$	2 μm
The width of the comb finger	3 μm

**Table 4 micromachines-14-00895-t004:** Parameters of the comb counter-pin actuator.

Parameter	Value
Comb height, $Z_{0}$	50 μm
Number of pairs of fingers, *N*	120
Length of the overlap of the combs, $X_{0}$	60 μm
Gap between the fingers, $Y_{0}$	2 μm
Width of the comb finger	3 μm

**Table 5 micromachines-14-00895-t005:** Design parameters of the elastic element.

Width, *b* μm	Length, *l* μm	Width, *h* μm
	Elastic beam $l_{1}$	
5	268	50
	Analytical stiffness, N/m	
	47.67	
	Elastic beam $l_{2}$	
5	258	50
	Analytical stiffness, N/m	
	42.535	

**Table 6 micromachines-14-00895-t006:** Parameters of the elastic suspension of the SE.

Parameter	Value
Material of the working layer	Si
Thickness of the working layer	50 μm
IM, $M_{1}$	1.6 × 10^−7^ kg
Weight of the electrostatic drive $M_{2}$	1.4 × 10^−8^ kg
Mass of measuring electrodes $M_{3}$	5.9 × 10^−9^ kg
Mass of the elastic element $M_{k}$	4.15 × 10^−10^ kg
Stiffness of the elastic element	21.736 N/m
Natural frequency in DM	4750 Hz
Natural frequency in SM	5000 Hz

**Table 7 micromachines-14-00895-t007:** The main parameters of the manufacturing process.

Parameter	Value
Wafer thickness, μm	450
Wafer resistivity, mΩ	15
Depth of the drawn alignment marks, μm	0.5
Thickness of the oxide (supports for SOI),	2
Depth of the cavity in the base, μm	80
Thickness of the working layer, μm	50
Resistivity of the working layer, mΩ	−
Thickness of the oxide above the working layer, μm	0.6
Thickness of the metallization layer (Al), μm	0.7
Specific surface resistance of the metallization layer (Al), $\Omega /m^{2}$	0.06
Depth of the “trench” for eutectic, μm	4
Width of the “trench” in the active structure, μm	2

**Table 8 micromachines-14-00895-t008:** List of topological layers of the manufacturing process.

Title	Layer Assignment	GDSII Layer Code
BASEMARK	Alignment labels	MRKB
SUPPORT	Supports for SOI	SUP
BASECAVITY	Cavities in the substrate	CAVB
ACTIVE	Active structures (movable mass)	ACT
METAL	Metallization layer	MET

**Table 9 micromachines-14-00895-t009:** The results of topological measurements of the SE.

Topological Element	Measured Value, μm	Nominal Value, μm	Measurement Accuracy, %
	**Linear dimensions of elements**		
Elastic beam length	269	270	−0.3
Length of the electrostatic actuator electrode	71.51	70	+2.1
Length of the finger of the capacitive converter	46.13	45	−1.8
Length of the locking converter	279.23	280	−0.28
Width of the U-shaped beam	4.8–5.17	5	±4
Width of the finger of the capacitive converter	2.88–3.00	3	−4/0
Width of the finger of the electrostatic actuator	2.7–2.9	3	−10/−3
Beam width of the electrostatic actuator	8.87	9	−1.44
	**Gaps between elements**		
Gap between the electrodes of the electrostatic actuator	2.38–2.5	2	+25
Gap between the electrodes of the capacitive converter	2.4–2.6	3	−13.3
Gap between the beams U-shaped elastic element	5.6–5.79	6	−6.7
	**Vertical size**		
Height of the structural layer	45...59	50	10–20

**Table 10 micromachines-14-00895-t010:** Results of the experiment to determine the amplitude–frequency response.

Parameter	DM	SM
External pressure	5000 Pa	5000 Pa
Constant voltage	+2	+1
Variable voltage	±5	±3
Scanning range	4700–4900 Hz	5000–5150 Hz
Natural oscillation frequency	4800 Hz	5060 Hz
Oscillation amplitude	1.646	1.041
Bandwidth level −3db	4.75 Hz	47.72 Hz
Q-factor	1010.5	106.03

## Data Availability

We did not report any data.

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
