# Peer review of "Development and Research of the Sensitive Element of the MEMS Gyroscope Manufactured Using SOI Technology"

_micromachines, 2023, doi:10.3390/mi14040895_

Round 1

Reviewer 1 Report

Thanks to the authors for the interesting material. However, I consider it necessary to make the following correction:

1. Line 72: sensing element 1 is not accurately shown.

2. Lines 122, 131, 621 and figures 2, 3: it is desirable to indicate the exact name of the specialized platform.

3. Lines 170-174: it is better to indicate the numbers in order. Or change the sequence of figures 4 and 5.

4. Figure 4: item 5 is missing (line 173).

5. Table 1: Please clarify whether the overlap area of the fingers is indicated with or without the extreme elements (Figure 6).

6. Lines 231-232: The authors chirp "As can be seen from the graph, it can be considered linear only in a small area from 0 um to 0.4 um...". Can you please clarify which part of the graph you mean?

7. Lines 258-259: The authors write "The most preferable will be a voltage of about 2 V, at which the force is 0.1 um". Please clarify the units of measurement with reference to Figure 9.

8. Line 272: how is the Kmm factor determined?

9. Line 328: In the explanation of Equation 11, the authors state "l is the length of the beam". This value is not present in this Equation.

10. Figures 15 and 18: Please indicate the parameters of the models shown in these figures.

11. Line 515: The authors write "...the same area can be calculated with a gap of 5 mm...". 5 mm is not a mistake?

12. Figure 25: why was the range from 0 to +280 degrees/sec used in the tests? More expected is the range from -280 to +280 degrees / sec when calibrating the microgyroscope.

Author Response

Dear Reviewer of MDPI Journal of Micromachines,
With this letter, on behalf of the team of authors Danil Naumenko, Alexey Tkachenko, Igor Lysenko and Andrey Kovalev from the Design Center of the Microelectronic Component Base for Artificial Intelligence Systems, Southern Federal University, Taganrog 347922, Russia. I would like to thank you for taking the time and opportunity to review the submitted manuscript and make constructive comments, which makes it possible to make the material of the submitted manuscript better. Below in this letter I would also like to give answers to your comments.
The manuscript submitted for review and subsequent publication is as follows:
"Development and Research of the Sensing Element of the MEMS Gyroscope Manufactured Using SOI Technology".
The subject of the manuscript submitted for consideration and publication is devoted to the consideration of the development, manufacture and research of an inertial angular acceleration sensor (MEMS gyroscope) based on MEMS technologies. Sent for subsequent publication in the MDPI Journal of Micromachines.

The team of authors believes that the manuscript submitted for consideration and subsequent publication fully corresponds to the subject and scientific directions of this journal, as well as the current world level of research and achievements in this scientific and technical field. The materials described in this manuscript can serve engineering and technical teams as a guide to the development and manufacture of angular acceleration sensors (MEMS gyroscope) based on MEMS technologies. The manuscript was written
and designed in accordance with the requirements for the design of scientific manuscripts of this journal. 

Responses to your comments:

1. Line 72: sensing element 1 is not accurately shown.
Answer: The indication of the sensor element (1) in Figure 1 has been corrected
(example MEMS gyroscope Tronics Gypro2300LD);

2. Lines 122, 131, 621 and figures 2, 3: it is desirable to indicate the exact name of the specialized platform.
Answer: The exact name of a specialized commercial platform for the development of inertial systems based on MEMS technologies and MEMS sensors is indicated. "Development Platform for MEMS Inertial Sensors". With a link to the source.

3. Lines 170-174: it is better to indicate the numbers in order. Or change the sequence of figures 4 and 5.
Answer: Figure 4 and Figure 5 - topology and cross-section of the developed
sensing element of the MEMS gyroscope have been changed to a more detailed and understandable one. Changed the indication and description of the main
design elements of the MEMS gyroscope. 

4. Figure 4: item 5 is missing (line 173).
Answer: Correction of this remark was made in the previous paragraph (3).

5. Table 1: Please clarify whether the overlap area of the fingers is indicated with or without the extreme elements (Figure 6).
Answer: Because of the cut fingers, the capacity decreases, the trimming is carried out in such a way as to be a multiple of the whole length of the comb. In this case, the adjustment is 2 fingers. This comment has been entered in the appropriate place in the manuscript, as well as some names in Table 1 have been corrected.

6. Lines 231-232: The authors chirp "As can be seen from the graph, it can be
considered linear only in a small area from 0 um to 0.4 um...". Can you please
clarify which part of the graph you mean?
Answer: It is worth considering that the graph is nonlinear, and the higher the
deviation, the greater its curvature. But with small deviations up to 0.3 um,
the deviation is not high and, to simplify calculations and detection, we take it
conditionally for linear.

7. Lines 258-259: The authors write "The most preferable will be a voltage of about 2 V, at which the force is 0.1 um". Please clarify the units of measurement with reference to Figure 9.
Answer: In this case, there was a gross typo. The units of measurement in this
case are measured in uN.

8. Line 272: how is the Kmm factor determined?
Answer: Typing errors. The variables in Equation (7) have been corrected. According to this Equation (7): it is necessary to subtract the electrostatic stiffness from the stiffness in drive mode (DM).

9. Line 328: In the explanation of Equation 11, the authors state "l is the length of the beam". This value is not present in this Equation.

Answer: Equation (11) l is the length of the elastic beam. The style of presentation of Equations and descriptions to them has been corrected.

10. Figures 15 and 18: Please indicate the parameters of the models shown in these figures.
Answer: For Figure 15, the geometric parameters of the model are shown in Figure 14 (geometric parameters of a U-shaped elastic beam). For Figure 18, the
geometric model is presented in Figure 17, however, a detailed description due to the large number of geometric parameters is not presented in this section of the manuscript.

11. Line 515: The authors write "...the same area can be calculated with a gap of 5 mm...". 5 mm is not a mistake?
Answer: Typing errors. Fixed assignment of 5 um.

12. Figure 25: why was the range from 0 to +280 degrees/sec used in the tests?
More expected is the range from -280 to +280 degrees / sec when calibrating the microgyroscope.
Answer: The graph shown in Figure 25 shows the dependence of the Coriolis force on the angular velocity, but not the range of the gyroscope developed by MEMS. When rotating in the other direction, the values will change the sign, but the modulus will be the same. In this case, we only need to estimate the magnitude modulus. The full range corresponds to: +/- 300 degrees / sec.

Reviewer 2 Report

Dear Authors, Thanks for submitting this review paper that addresses a new method to develop MEMS Gyroscopes. I believe the paper does not cover a wide range of related studies. There are a lot of studies in this field. A minimum of 70 references is needed. This is mandatory in order to compare the value of this work to those already published. Tables need more information specially for the overlapping point. Some of the symbols appear in the text do not appear in the equation. Consistency is an issue. Writing style needs improvement.

Thank you

Author Response

Dear Reviewer of MDPI Journal of Micromachines,

With this letter, on behalf of the team of authors Danil Naumenko, Alexey Tkachenko, Igor Lysenko and Andrey Kovalev from the Design Center of the Microelectronic Component Base for Artificial Intelligence Systems, Southern Federal University, Taganrog 347922, Russia. I would like to thank you for taking the time and opportunity to review the submitted manuscript and make constructive comments, which makes it possible to make the material of the submitted manuscript better. Below in this letter I would also like to give answers to your comments.
The manuscript submitted for review and subsequent publication is as follows:
"Development and Research of the Sensing Element of the MEMS Gyroscope Manufactured Using SOI Technology".
The subject of the manuscript submitted for consideration and publication is devoted to the consideration of the development, manufacture and research of an inertial angular acceleration sensor (MEMS gyroscope) based on MEMS technologies. Sent for subsequent publication in the MDPI Journal of Micromachines.
The team of authors believes that the manuscript submitted for consideration and subsequent publication fully corresponds to the subject and scientific directions of this journal, as well as the current world level of research and achievements in this scientific and technical field. The materials described in this manuscript can serve engineering and technical teams as a guide to the development and manufacture of angular acceleration sensors (MEMS gyroscope) based on MEMS technologies. The manuscript was written
and designed in accordance with the requirements for the design of scientific manuscripts of this journal.
The manuscript submitted for consideration for publication and the materials described in it are undoubtedly of practical importance in the development of scientifically based solutions for the design and manufacture of MEMS gyroscopes, methods and means of studying their characteristics. 

Responses to your comments:

- Corrections in the manuscript being sent and considered for publication after reviewing touched upon the general style and sequence of writing the manuscript, input errors were corrected (errors in equations, variables in equations, signatures to equations, sections and subsections, writing tables, figures and explanations to figures).
- The list of literary sources has been expanded constructively to cover the area of knowledge under study, including a larger number of international scientific papers.

On behalf of the team of authors of the article, I express the hope that our responses to your comments and the changes made to the manuscript submitted for review and publication will be acceptable and correct.
We are also ready to additionally answer all your questions related to the material of the manuscript and this study, not in the format of a review, but via e-mail, for example. We are interested in maintaining cooperation and exchange of experience between research and engineering teams at any level. 

Sincerely, the scientific team of the authors of the manuscript of the
Southern Federal University, Design Center of the Microelectronic Component Base for Artificial Intelligence Systems
Danil Naumenko <[email protected]>
Alexey Tkachenko <[email protected]>
Igor Lysenko <[email protected]>
Andrey Kovalev <[email protected]>

Round 2

Reviewer 2 Report

Thank you for the update version.